# Human multiethnic radiogenomics reveals low-abundancy microRNA signature in plasma-derived extracellular vesicles for early diagnosis and molecular subtyping of pancreatic cancer

Jianying Xu[1], Wenjie Shi[2†], Yi Zhu[3†], Chao Zhang[4†], Julia Nagelschmitz[2], Maximilian Doelling[2], Sara Al-Madhi[2], Ujjwal Mukund Mahajan[1], Maciej Pech[5], Georg Rose[6], Roland Siegfried Croner[2], Guoliang Zheng[7*‡], Christoph Kahlert[8*‡], Ulf Dietrich Kahlert[2,6*‡]

[1]Department of Medicine II, Hospital of the LMU, Munich, Germany; [2]Molecular and Experimental Surgery, Clinic for General-, Visceral -, Vascular- and Transplantation Surgery, Medical Faculty and University Hospital Magdeburg, Otto-von-Guericke University, Magdeburg, Germany; [3]Department of Gastroenterological Surgery, The Affiliated Hospital of Jiaxing University, Jiaxing, China; [4]Department of Radiology, The First Affiliated Hospital of Wannan Medical College, Wannan, China; [5]Clinic for Radiology and Nuclear Medicine, University Hospital Magdeburg, Magdeburg, Germany; [6]Research Campus Stimulate, Otto von Guericke University Magdeburg, Magdeburg, Germany; [7]Department of Gastric Surgery, Cancer Hospital of China Medical University (Liaoning Cancer Hospital and Institute), Shenyang, China; [8]Department of General, Visceral and Transplantation Surgery, Heidelberg University Hospital, Heidelberg, Germany

*For correspondence:
zhengboren1@126.com (GZ);
christoph.kahlert@med.uni-heidelberg.de (CK);
ulf.kahlert@med.ovgu.de (UDietrichK)

†These authors contributed equally to this work
‡These authors also contributed equally to this work

Competing interest: The authors declare that no competing interests exist.

## eLife Assessment

The authors attempt to identify which patients with benign lesions will progress to cancer using a liquid biomarker. Although the study is **valuable**, the evidence provided for the liquid biopsy EV miRNA signature developed based on radiomics features remains **incomplete**. There remain key details missing and validation experiments that would better support the conclusions of the study.

**Abstract** Pancreatic cancer (PC) is a highly aggressive malignancy in humans, where early diagnosis significantly improves patient outcomes. However, effective methods for accurate and early detection remain limited. In this multiethnic study involving human subjects, we developed a liquid biopsy signature based on extracellular vesicle (EV)-derived microRNAs (miRNAs) linked to radiomics features extracted from patients' tumor imaging. We integrated eight datasets containing clinical records, imaging data of benign and malignant pancreatic lesions, and small RNA sequencing data from plasma-derived EVs of PC patients. Radiomics features were extracted and analyzed using the limma package, with feature selection conducted via the Boruta algorithm and model construction through Least Absolute Shrinkage and Selection Operator regression. Radiomics-related low-abundance EV miRNAs were identified via weighted gene co-expression network analysis and validated for diagnostic accuracy using 10 machine-learning algorithms. Three key EV miRNAs were

found to robustly distinguish malignant from benign lesions. Subsequent molecular clustering of these miRNAs and their predicted targets identified two PC subtypes, with distinct survival profiles and therapeutic responses. Specifically, one cluster was associated with prolonged overall survival and higher predicted sensitivity to immunotherapy, while the other indicated high-risk tumors potentially amenable to targeted drug interventions. This radiogenomic EV miRNA signature in human plasma represents a promising non-invasive biomarker for early diagnosis and molecular subtyping of PC, with potential implications for precision treatment strategies.

## Introduction

Pancreatic cancer (PC) is one of the most lethal tumors having a poor prognosis, and patients suffering from this disease show one of the lowest 5-year overall survival rates of all cancer patients, with approximately 13%. One of the main reasons for this dismissal prognosis is the lack of a proper early detection possibility, resulting in late diagnosing often in advanced, metastatic stage (*Bamankar and Londhe, 2023*).

The detection and diagnosis of PC, of which approximately 90% are classified as pancreatic ductal adenocarcinoma (PDAC), currently relies primarily on modalities of medical imaging, such as computed tomography (CT), magnetic resonance imaging, positron emission tomography, and transabdominal ultrasonography (*Wang et al., 2021*). The most common biomarker considered for PDAC differential diagnosis is elevated blood abundance of carbohydrate antigen 19-9 (CA19-9) and carcinoembryonic antigen, though these are only used as prognostic markers and not effective for screening or early diagnosis (*Bestari et al., 2024*). Nowadays, new markers based on liquid biopsy such as microRNA are discerned and might pose as promising tools for early detection of PDAC. MicroRNAs (miRNAs) are non-coding RNAs that target genes and regulate their expression by inhibiting mRNA translation or enhancing their degradation (*Rashid et al., 2024*). Currently, extracellular vesicles (EVs) are gaining attention as disease-specific markers since they carry the material of their secreting cells and are therefore considered to contain tumor-derived elements, showcasing their molecular fingerprint (*Bamankar and Londhe, 2023*). It has been shown multiple times that miRNA derived from small EVs play a role in differentiation and metastasis of cancer (*Mok et al., 2024*; *Min et al., 2019*). The rapidly developing field of data mining and analytical techniques provides new insights and makes the discovery of relevant key players more feasible. Numerous miRNAs have been described using co-expression network analysis that might be applied as diagnostic or prognostic biomarker, for patient stratification or disease recurrence (*Tiwari et al., 2024*).

Benefiting from interdisciplinary advances in artificial intelligence, the integration of machine learning and genomics has led to breakthroughs in the early diagnosis and classification of tumors. For example, we used machine-learning algorithms to assist in the development of a three-serum miRNA signature that effectively provides early warning of premalignant PC (*Shi et al., 2024*). Another model, based on machine-learning algorithms, focuses on the immune subtypes of triple-negative breast cancer, offering critical insights for identifying patients who may benefit from immunotherapy (*Chen et al., 2021*). In the current study, we employed radiogenomics technology, another product of interdisciplinary collaboration between medicine and engineering. This novel approach integrates the quantification of image features from CT or magnetic resonance imaging, which are then correlated with genomic signatures and allows for a non-invasive prediction of molecular characteristics (*Casà et al., 2022*). Such as, claim to be able to predict the occurrence of p53 mutations using CT images by radiogenomic analysis and hence make a statement on the prognosis (*Iwatate et al., 2020*).

In the present study, we aim to develop a liquid biopsy signature of EV-derived miRNA based on the radiogenomic analysis of CT images derived from ethically diverse backgrounds and interrogating small RNA-seq data of plasma-derived total EVs, in order to advance the diagnostic possibilities and the molecular subtyping of PDAC.

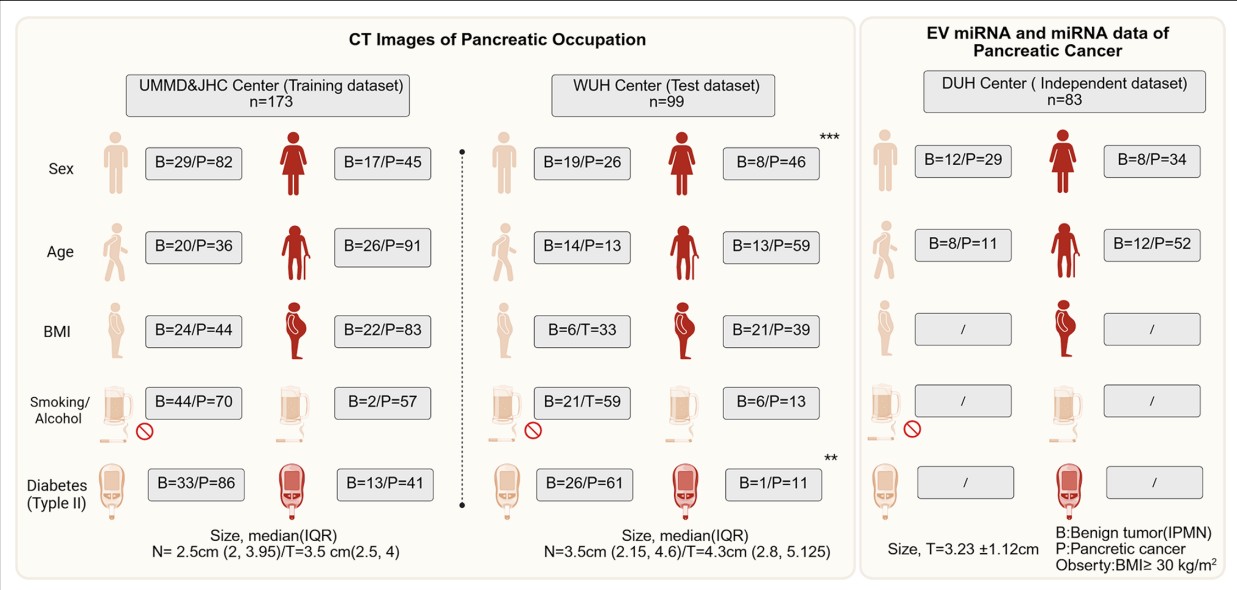

**Figure 1.** The baseline information of clinical parameters of patients enrolled from four centers in this multicenter trial. The intergroup comparisons were performed using the chi-square test. The * means p less than 0.05, ** means p less than 0.01, and ***means p less than 0.001.

## Results

### Enrolled population and baseline information for CT images and EV miRNA

A total of 272 patients enrolled in this study providing CT imaging, including 173 in UMMD and JHC center while 99 in WUH center. In the UMMD and JHC center, also including 46 pancreatic benign lesion and 127 PC CT images. Most pancreatic patients in this center are older with obesity, but less smoking or alcohol and less with diabetes. For the WUH center, most PC patients are female and also have a high incidence of older age and obesity. The DUH provides the patients with EV miRNAs, EV mRNAs data, and follow-up information. About 82.5% (52/63) patients are older, and 34 patients are female. The clinical characteristics of the data cohorts are summarized in *Figure 1*.

### Different expression radiomic features between pancreatic benign lesions and aggressive tumors

Before the analysis, we conduct the propensity score matching (PSM) procedure to match the benign and aggressive tumor from DUH to JHC, respectively, according to age. After the match, we found both center baseline differences were removed (*Figure 2A*). Then the difference expression radiomics features was conducted, the results indicate that a total of $n = 88$ significant features demonstrate differences between groups (*Figure 2B*).

### Four important radiomic features were selected to build a related signature

We use Boruta algorithms to select the important features, and the results show that a total of 12 features were identified (*Figure 2—source data 1*), which is more than shadow features. In addition, after inputting the above features into Lasso regression (LR) algorithms, we found that four features are listed as key features (*Figure 2D*). Based on the LR model with regression coefficient and feature expression, we build a four radiomics feature-related signature and validate the prediction ability in WUH center data, revealing a signature accuracy of prediction efficiency (area under the curve [AUC] = 0.911) (*Figure 2E, F*).

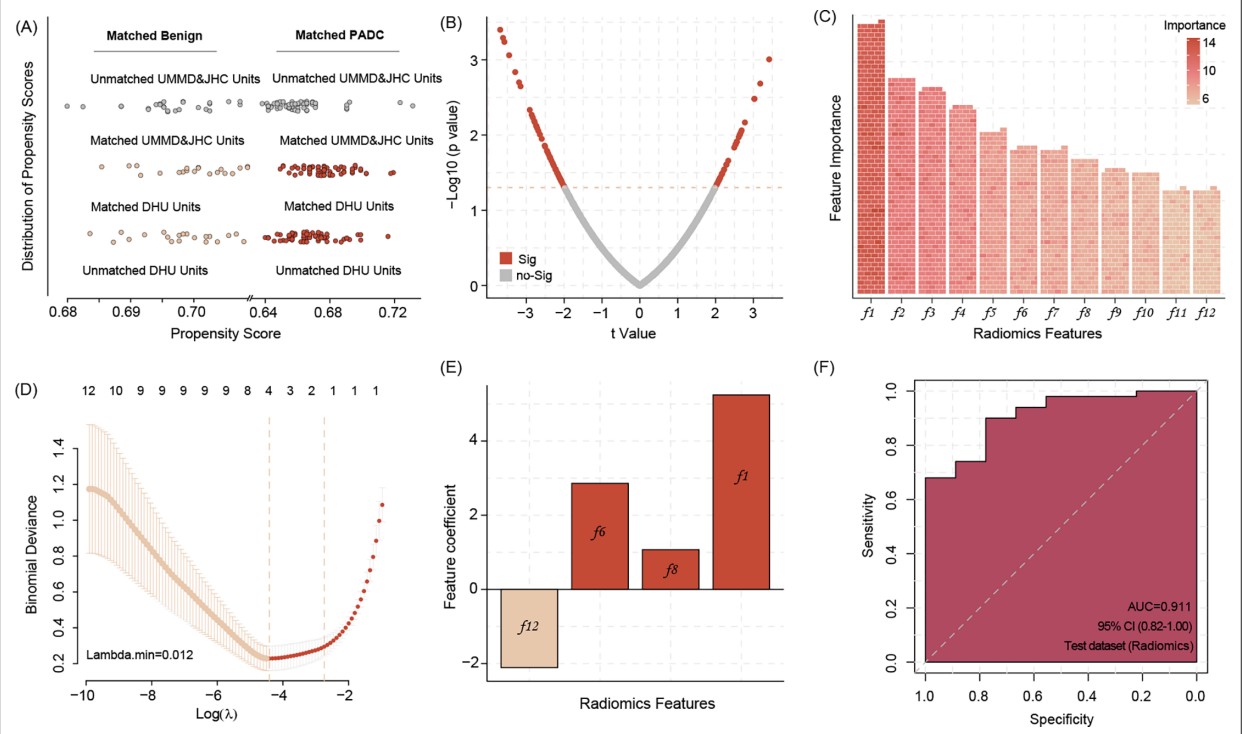

**Figure 2.** Propensity score matching (PSM) allows matching data of benign pancreas lesions and pancreatic ductal adenocarcinoma (PDAC) patients from DUH to UMMD and JHC patients according to the age factor, all of DUH patients successfully matched similar patients (**A**). The different radiomic features between the benign lesions and PDAC patients' computed tomography (CT) images (**B**) Twelve most important radiomic features differentiating between the benign pancreatic lesion and PDAC patients' CT images identified by the Boruta algorithms (**C**) Four radiomic features were selected by Least Absolute Shrinkage and Selection Operator (LASSO) regression to build model signature (**D, E**). Applying the four radiomic features related signature in image analysis shows high accuracy in predicting the PDAC manifestation in the WUH test dataset (**F**).

The online version of this article includes the following source data for figure 2:

**Source data 1.** Important radiomic features.

### Three EV miRNAs are associated with radiomic features

After the radiomics signature was built, each patient presented an individual risk score, and we split patients into high- and low-risk patients, according to the median value of risk score. We use weighted gene co-expression network analysis (WGCNA) to connect the EV miRNA data and two imaging-featured patient risk groups. According to the WGCNA analysis, the green module was identified as the key module and includes 12 hub co-expressed miRNAs. The number of low-abundance miRNAs calculated was $n = 295$. Merging both results, three miRNAs are identified (hsa-miR-1260b, hsa-miR-151a-3p, and hsa-miR-5695) and selected for subsequent alignment with associated radiomic features (*Figure 3A–C*).

### Expression validation of three EV miRNAs

We use serum and tissue samples to validate the expression of three hub EV miRNAs, and the results show that compared with healthy serum samples, these three miRNAs are enriched in tumor patient serum samples. Interestingly, this correlation of upregulation in tumor conditions was also true when comparing tumor tissue with non-tumor tissue (*Figure 3D–I*).

### Three EV miRNAs predicted PC with high accuracy

We use seven machine-learning combos to train and test the ability of three EV miRNA levels to predict tumor manifestation and clinical course of patient. The results show that in the GBM-default (cut-off = 0.75) model, three EV miRNAs show a high accuracy to predict the PC with the training accuracy of 0.978 with AUC = 0.978, and two test dataset accuracies are 0.923 with AUC = 0.919, and 0.871 with AUC = 0.857, respectively. Then, we choose the GBM model for the extended validation

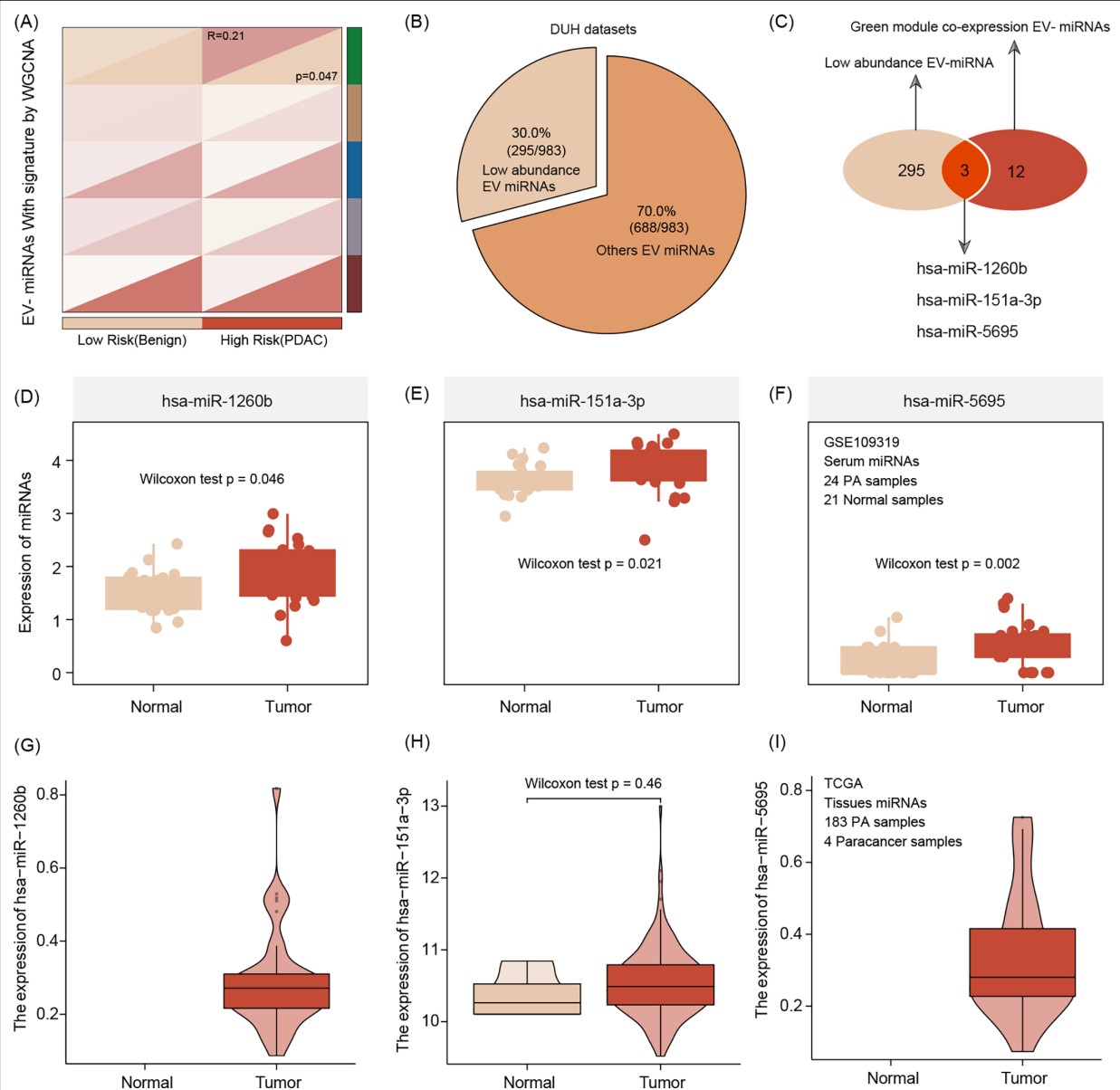

**Figure 3.** Extracellular vesicle (EV) miRNA presenting the risk group stratification based on radiomics signature by weighted gene co-expression network analysis (WGCNA) featuring green mode discovering our key module for further analysis (*r* = 0.21, *p* = 0.047).(**A**). The number of low-abundance miRNAs in the entire EVseq dataset cohort is *n* = 295 (**B**) Out of those low-abundance miRNAs, *n* = 12 present matching candidates differentially expressed in high-risk group patients. Alignment to our radiomics feature parameters identified three core miRNAs (hsa-miR-1260b, hsa-miR-151a-3p, and hsa-miR-5695) (**C**) The three key miRNAs show significantly different expression levels in tumor conditions, both for serum (**D–F**) and tissue (**G–I**) The Wilcoxon test was used to compare differences between two groups.

by our hospital data (MUH) and GSE109319. Before this procedure, we use the combat package to remove the batch effect allowing the merge of the two datasets. The results of the extended dataset validation via GBM further highlight the high diagnostic accuracy of the three EV miRNAs (accuracy = 0.894, and AUC = 0.897) (*Figure 4*), while the CA19-9 predict AUC is 0.843 (95% CI, 0.762–0.924).

## Identification of two molecular subtypes of PDAC with significant clinical predictive value

After predicting molecular targets of three miRNAs, we obtain a number of 117 mRNA shared candidate targets (*Figure 5—source data 1*). Stratifying the tumors according to the level of expression of

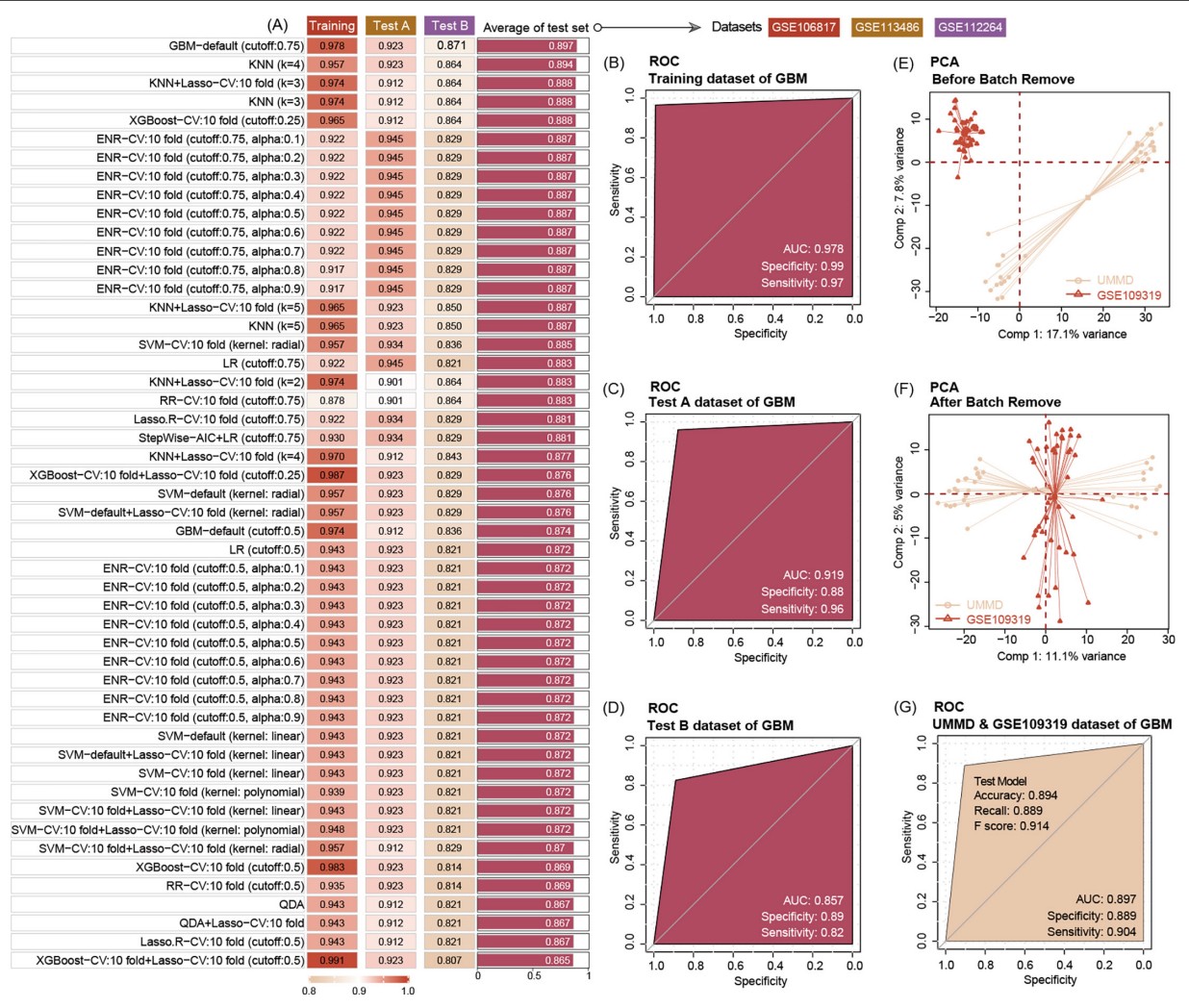

**Figure 4.** Ten machine-learning algorithms demonstrate that three key miRNAs show a high accuracy to diagnose pancreatic ductal adenocarcinoma (PDAC) in early stage, no matter in training or test datasets. The best machine-learning algorithms are GBM (cut-off: 0.75) (**A**). Three miRNAs' prediction ability of the training dataset (GSE10106817) in the GBM model is 0.978 (**B**). Three miRNAs' prediction ability of the test dataset (GSE113486 and GSE112264) in the GBM model is 0.919 and 0.857, respectively (**C, D**). Data distribution before removal of batch effect of our center data and GSE109319 dataset (**E**). Data distribution after removing batch effect of our center and GSE109319 dataset (**F**). Three extracellular vesicle (EV) miRNAs' prediction ability to identify cancer of our center data and GSE109319 in GBM model is 0.897 (**G**).

those allows the differentiation of tumors in the cluster into two subtypes termed Cluster 1 (C1) and C2 (*Figure 5—source data 2*). The survival analysis shows that compared with C1, the C2 patients feature a prolonged survival outcome, no matter in overall survival time (OS) or disease free survival time (DFS) (*Figure 5B, C*). In addition, analyzing the relationship with clinical factors of patients, we also found the C1 subtype patients are older in age and carry bigger tumor sizes and higher number of tumor cell positive lymph nodes (*Figure 5D–F*), when compared with C2 subtype patients. We also demonstrate C1 patients are particularly aggressive in female patients, featuring a high number of tumors containing perineural invasion (PNI) and advanced pathological tumor staging (*Figure 5G–I*).

## C2 patients with high immune infiltration and positive with immune checkpoints

Testing for gene signatures that indicate immune cell existence, we show that C2 tumors are significantly more enriched for immune cell signals, including those for CD8 T cells, cytotoxic lymphocytes, and NK cells (*Figure 6A, B*). In addition, samples of patients of that subtype also show a more physiological expression signature in terms of immune checkpoints as compared to C1 tumors (*Figure 6C*).

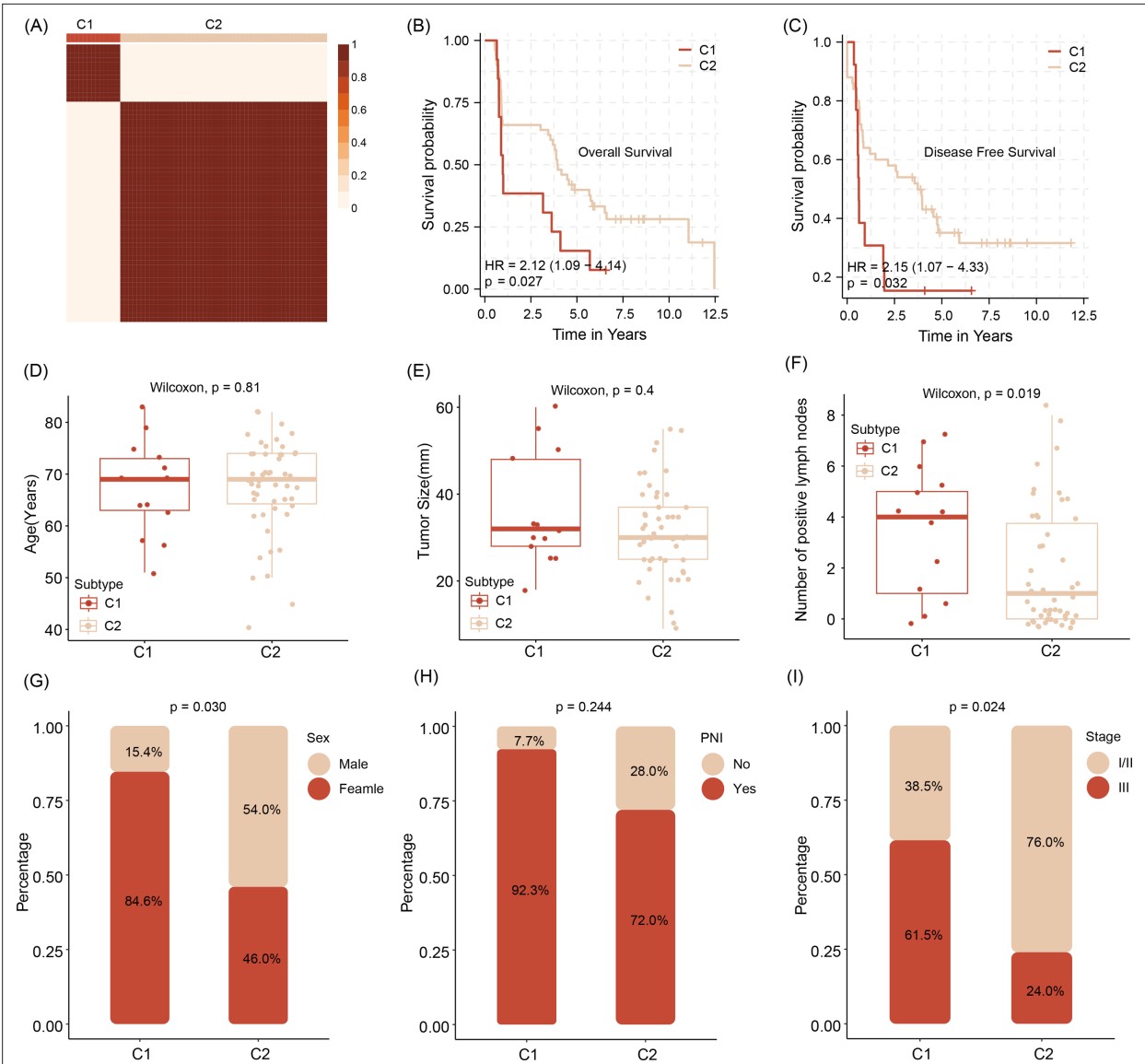

**Figure 5.** Stratification of abundancy levels of shared mRNAs by non-negative matrix factorization method allows the clustering of patients into two subtypes (C1 and C2) (**A**). Patients of the C1 with a poor outcome in OS and DFS (**B, C**). C1 subtype patients are characterized by older age and bigger average tumor size, higher number of tumor cell positive lymph nodes as compared to C2 patients (**D–F**). C1 patients are predominantly female, have tumors with pathological classification marks of perineural invasion/ PNI and advanced tumor stage (**G–I**).Survival analysis was performed using the log-rank test. Differences in continuous variables between groups were assessed using the Wilcoxon rank-sum test, while differences in categorical variables were analyzed using the chi-square test.

The online version of this article includes the following source data for figure 5:

**Source data 1.** Shared targets of three extracellular vesicle (EV) miRNAs.

**Source data 2.** NMF rank survey of shared targets of three extracellular vesicle (EV) miRNAs.

## Prediction of drug sensitivity of C1 patients

We use Genomics of Drug Sensitivity in Cancer (GDSC) data to merge cell line response data and expression data with expression signals in C1 and C2 tumors. This analysis indicated that C1 patients may benefit from therapy with FDA-approved AKT inhibitor VIII, Bleomycin, Dasatinib, GNF-2, PF-562271, Refametinib, and BMS-509744 (*Figure 6D*).

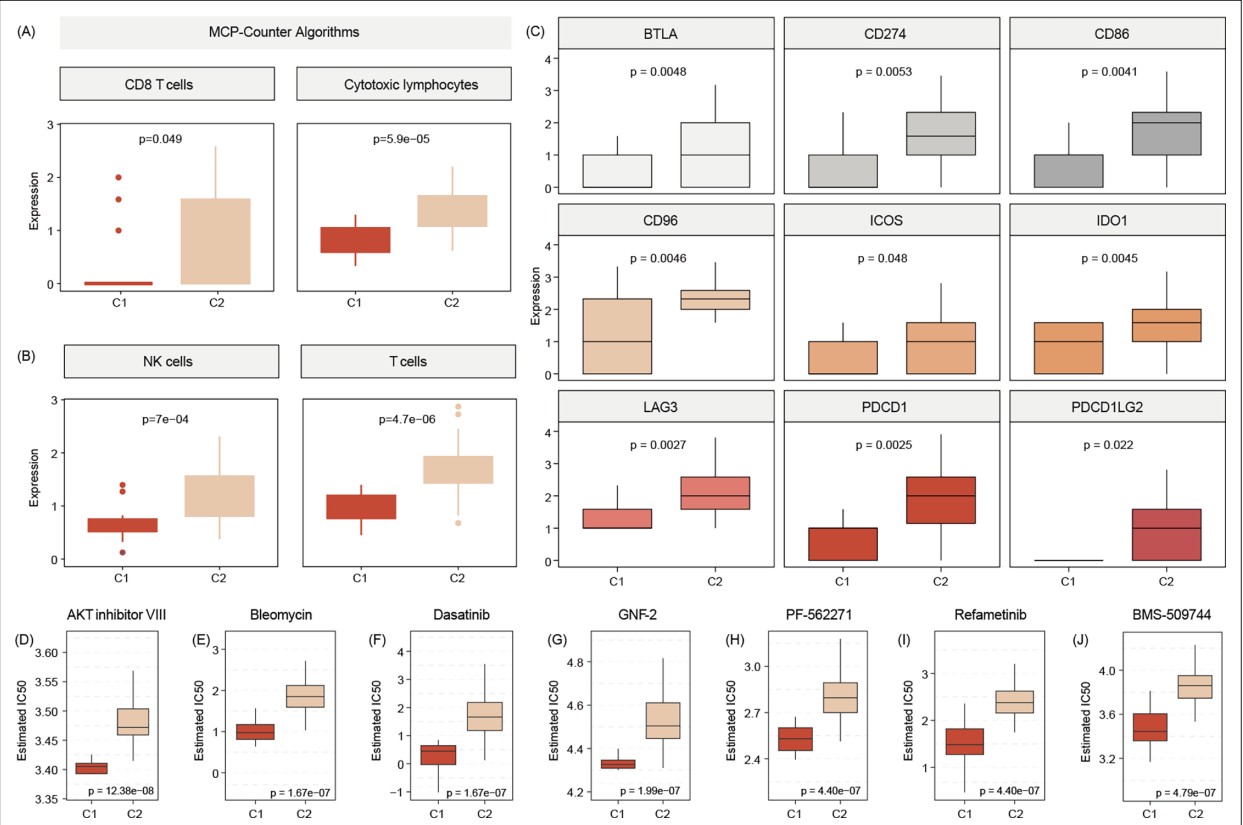

**Figure 6.** C2 subtype is positively associated with elevated levels of transcripts regulating gene pathways encoding for CD8 T cells, cytotoxic lymphocytes, and NK cells (**A, B**). Commonly known immune checkpoints are also higher expressed in C2 subtypes (**C**) Aligning in vitro drug sensitivity and expression data from the GDSC database to subtype signature predicts that tumors of C1 patients might be more sensitive to AKT INHIBITOR VIII, Bleomycin, Dasatinib, GNF-2, PF-562271, Refametinib, ans BMS-509744 as C2 subtypes (**D-J**).The Wilcoxon test was used to compare differences between two groups.

## Functional annotation and pathway enrichment for the C1 subtype

GO functional enrichment analysis indicated that the C1 subtype is enriched for intermediate filament organization, GOCC ribosome, as well as GOMF symporter activity (*Figure 7A, B*). Pathway enrichment analysis showed that the C1 subtype was activated with the Reactome fatty acids, Reactome diseases of metabolism, as well as Reactome biological oxidations pathways, and may be inhibited with Reactome apoptosis, Reactome DNA repair, and Reactome signaling by Hippo pathways (*Figure 7C*).

## Discussion

Using multicenter radiomics of Asian and Western world patients, we identified three plasma total EV fraction microRNAs that demonstrated strong predictive performance and prognostic value, offering new insights for non-invasive early diagnosis and prognosis of PC. This study is the first to distinguish between benign lesions and PC by identifying novel and significant plasma EV microRNAs based on differential radiomics features. We focused on low-abundance microRNAs, which are often overlooked in routine RNA sequencing analyses. These three microRNA candidates were validated across multiple cohorts, achieving excellent predictive performance in both testing and external validation sets. Additionally, these microRNAs demonstrated prognostic value, potentially aiding in treatment selection.

PC remains one of the most lethal diseases, with insufficient early detection methods and limited treatment options (*Halbrook et al., 2023*; *Fahrmann et al., 2021*). This study aimed to develop a liquid biopsy nucleic acid diagnostic test based on EV-derived microRNA using radiogenomic analysis

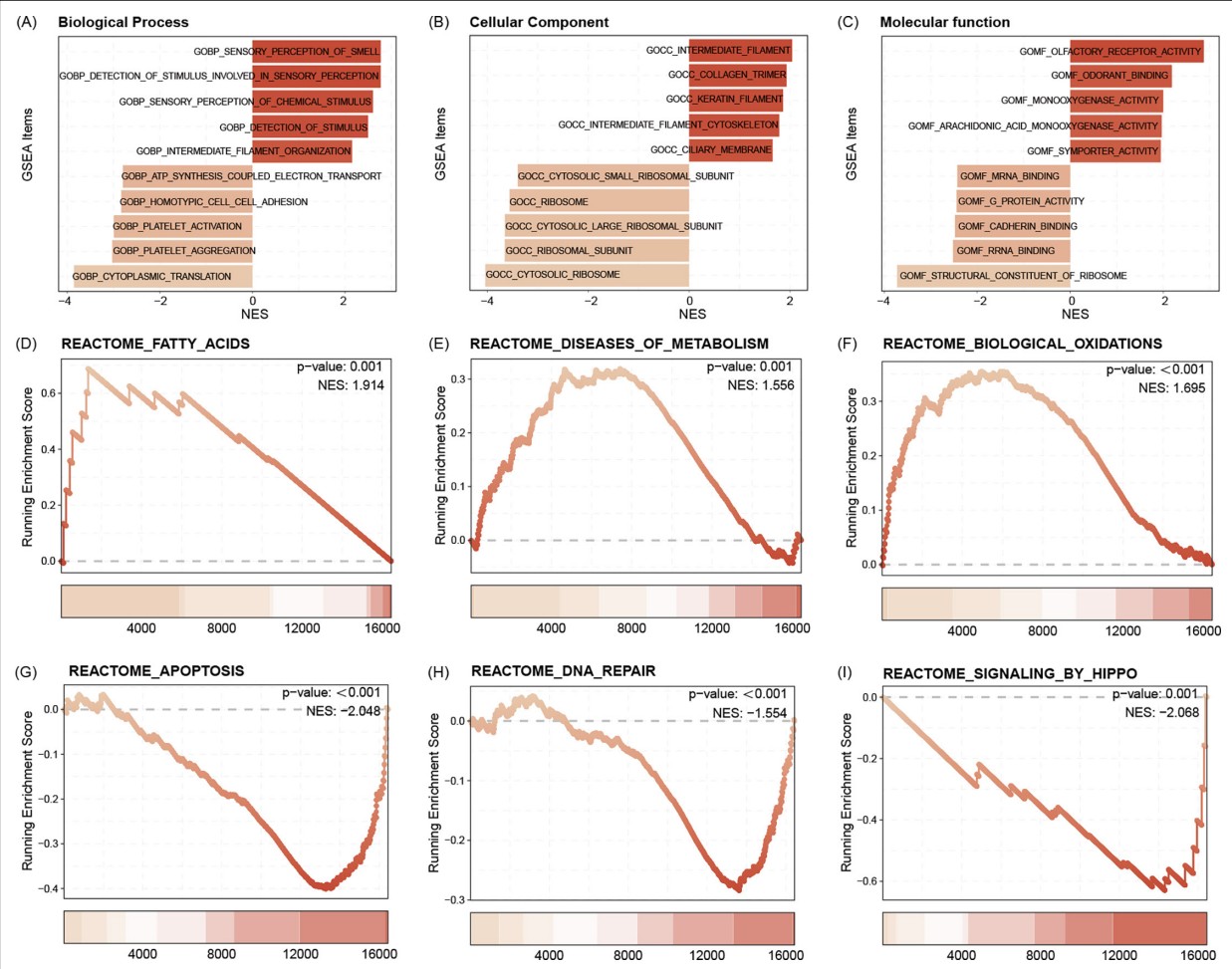

**Figure 7.** GO functional enrichment analysis indicated that the C1 subtype is enriched for intermediate filament organization, CC ribosome, as well as MF symporter activity (**A–C**). Pathway enrichment analysis showed that the C1 subtype was activated with the Reactome fatty acids, Reactome diseases of metabolism, as well as Reactome biological oxidations pathways, and may be inhibited with Reactome apoptosis, Reactome DNA repair, and Reactome signaling by Hippo pathways (**D–I**).

to enhance diagnostic accuracy and molecular subtyping of PDAC. By focusing on imaging features to distinguish benign lesions from PC, we identified underlying genetic factors contributing to these differences. Unlike previous studies on serum microRNA (*Wei et al., 2020*; *Shams et al., 2020*), our research identified three microRNAs through radiomics that reflect tumor-specific changes rather than systemic variations. Additionally, our findings reveal relationships between imaging features and tumor biology, improving diagnostic accuracy and offering a more comprehensive tool for early detection and personalized disease management.

Although numerous studies have reported on microRNA sequencing results (*Yang et al., 2021*), highly abundant microRNAs often mask the signals of many others that are present at very low abundance during routine analysis. Furthermore, if a microRNA is associated with tumor development, it is likely to be in low abundance during early detection and may even be undetectable in benign lesions. In our study, after identifying 12 microRNAs related to radiomic differences, we further defined three (hsa-miR-1260b, hsa-miR-151a-3p, and hsa-miR-5695) of them as being of low abundance. These microRNAs showed significantly higher expression in PC blood than in non-malignant cancer patients' plasma, with hsa-miR-1260b and hsa-miR-5695 undetectable in PC samples. The model based on these three microRNAs demonstrated strong performance in the training group (AUC, 0.978; sensitivity, 0.97; specificity, 0.99), testing group (AUC, 0.919; sensitivity, 0.96; specificity, 0.88), and external testing group (AUC, 0.897; sensitivity, 0.90; specificity, 0.89).

Previous studies on microRNAs as diagnostic biomarkers for PC vs healthy controls report AUCs of 0.78–0.96 and sensitivities of 0.62–0.88 (*Yang et al., 2021*), but many lack external validation (*Liu et al., 2012*; *Wang, 2020*). One case–control study identified two diagnostic panels based on microRNA expression (index I: 4 microRNAs; AUC, 0.86; sensitivity, 0.85; specificity, 0.64; index II: 10 microRNAs; AUC, 0.93; sensitivity, 0.85; specificity, 0.85) (*Schultz et al., 2014*). Our study achieved similar performance using only three microRNAs, making it more practical for clinical application.

To the best of our knowledge, hitherto, none of the three identified microRNAs have been reported in the context of diagnostic biomarkers in PC. Moreover, to our knowledge, hsa-miR-5695 has only indirectly been linked to breast carcinogenesis (*Søkilde et al., 2019*) and subtypes of metastatic prostate cancer (*Watahiki et al., 2011*), while hsa-miR-1260b has been implicated in promoting tumorigenesis in lung cancer (*Kim et al., 2021*), breast cancer (*Park et al., 2022*), sarcoma (*Morita et al., 2020*), and prostate cancer (*Hirata et al., 2014*). Interestingly, in a screen of microRNAs that are differentially expressing in total plasma fraction of patients suffering from intraductal papillary mucinous neoplasm (IPMN) as compared to non-diseased controls, hsa-miR-1260b was identified as one of the top upregulated signals in plasma of IPMN cases (*Permuth-Wey et al., 2015*). Our work further enforces the potential of hsa-miR-1260b abundance in plasma to detect onset of abnormal pancreas genesis. Exosomal hsa-miR-151a-3p has emerged as a potential novel biomarker for predicting bone metastasis in lung cancer (*Zhao et al., 2023*). A very recent report of results of a multicenter trial in Asian participants also identified circulating 2′-O-methylated form of hsa-miR-151a-3p is upregulated in total fraction of plasma of PDAC patients compared to those derived from healthy donors or patients with chronic pancreatitis (*Yang et al., 2024*). Focusing on EV fraction, our results verify the potential of this newly discovered biomarker in PC blood diagnosis.

PC currently has only two standard systemic treatments with limited efficacy, and the patient population responsive to immunotherapy and targeted therapy remains unclear (*Neoptolemos et al., 2018*). Therefore, we explored the prognostic value of these microRNAs, which allowed for patient stratification. Patients classified as C2 had better prognoses, with higher immune cell infiltration and more immune checkpoint expression, suggesting they may be suitable for immunotherapy after standard treatment. In contrast, patients classified as C1 had poorer prognoses, related to inhibited pathways such as DNA repair and apoptosis.

The findings suggest potential for developing kits for early detection and treatment stratification and highlight these three candidates for further exploration. However, despite thorough matching of imaging, genomic, and clinical data, biases may still exist. Additionally, using the Microenvironment Cell Populations-counter (MCP-counter) algorithm for blood samples is a limitation of our study, as it was originally designed for solid tissues, which may affect the accuracy of cell population quantification in the blood. Furthermore, when validating the three microRNAs in tissue, the sample size of paracancerous tissue was limited. Lastly, since our EV purification did not use measures to enrich for cancer cell-derived EVs such as CD147 membrane presenting amplitude (*Ko et al., 2023*), we cannot exclude contamination of our miRNA data due to signals from non-cancer cell background, which might impact the cancer specificity of our proposed blood test. Moreover, our data is based on OMICS data acquisition. Real-world application of our proposed test would benefit from proofing the predictability and sensitivity of the miRNAs by targeted methods, such as RT-qPCR or CRISPR/Cas diagnostics and in prospective, multicenter trials. In addition, we acknowledge the absence of third-phase validation results, which will limit application in the clinical practice.

## Materials and methods
### Data resources and pre-analytics preparation
A total of four hospitals and four public datasets, comprising a total of eight datasets enrolled in this study, University Hospital Magdeburg in Germany (UMMD), and Jiaxing Hospital Center, China (JHC), provided enhanced CT images of 46 IPMN patients and 127 PC patients as training dataset. For test datasets, CT data resource from Wanan Medical University Hospital (WUH) with 27 PB patients and 72 PC patients. University Hospital Dresden, Germany (DUH) center provides miRNA and mRNA seq data from total plasma EVs of PDAC patients with associated clinical follow information, including 20 benign pancreatic disease patients and 63 PC patients. The four public serum miRNA sequence data containing PC and healthy control (HC) were GSE106817 (2759 PC vs.115

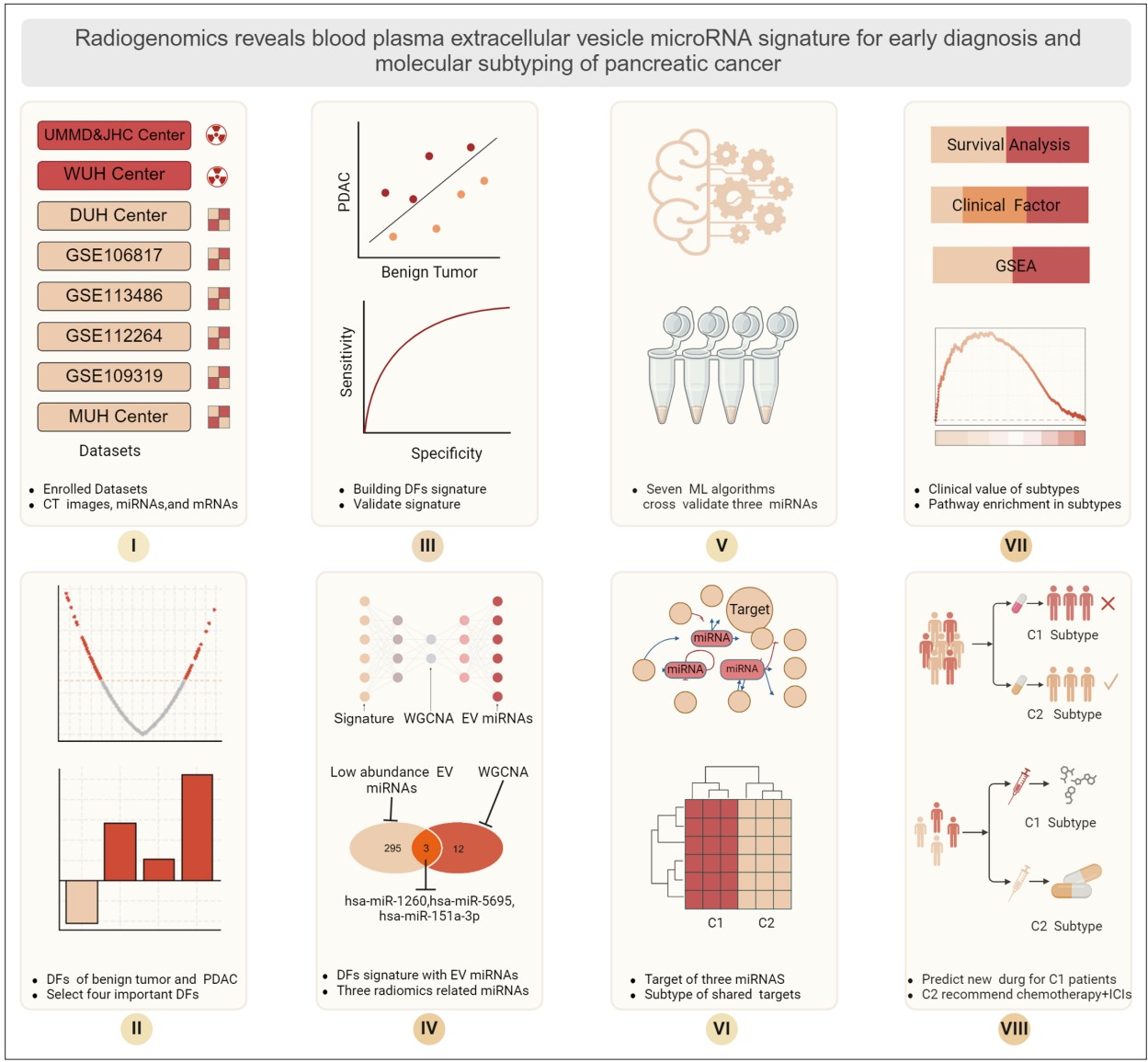

**Figure 8.** Schematic presentation of the workflow of this study. The use of radiomics to aid finding novel extracellular vesicle (EV)-charged miRNAs to allow pancreatic ductal adenocarcinoma (PDAC) diagnostics.

HC), GSE113486 (40 PC vs.100HC), GSE112264 (50 PC vs.41 HC), and GSE109319 (24 PC vs. 21 HC). All CT images were acquired using a 64-slice multidetector spiral CT scanner, with a standard slice thickness of 1.0–1.5 mm and a reconstruction interval of 1 mm. All PC patients underwent a standard pancreatic protocol triphasic contrast-enhanced CT examination. For radiomics analysis, images from the portal venous phase were selected due to their consistent clarity in delineating pancreatic tumor boundaries and surrounding vasculature. To ensure data consistency, all imaging data underwent preprocessing, including resampling, intensity normalization of grayscale values, and N4 bias field correction to address potential low-frequency signal inhomogeneities. We use the PSM method to match the benign and tumor patients from DUH to UMMD and JHC according to age factor. Afterward, we constructed a new matrix including CT images, EV miRNAs, mRNAs, and patients' clinical follow-up information. The workflow is schematically depicted in *Figure 8*. Ethical approval to conduct the study for UMMD and DUH after approval by the local Institutional Review Board/ethics committee (UMMD 46/22; 30/01 with amendment 43/14; DUH: EK76032013). Written informed consent from the patients was obtained pre-operatively with the disclosure of research purpose.

## EV isolation and RNA sequencing

Details on the protocols for EV isolation including presentation of optical and molecular characteristics of isolated vesicles proofing high-quality isolation performance have been described previously by our groups (*Han et al., 2023*).

EV isolation using ultracentrifugation: 500 µl plasma samples were thawed and mixed with 500 µl phosphate-buffered saline (PBS). The diluted plasma samples were filtered with 0.2-µm filter and subjected to ultracentrifugation at $100,000 \times g$, 2 hr, 4°C in a ultracentrifuge (Sorvall MX150 + micro-ultracentrifuge, Thermo Scientific, Darmstadt, Germany). The supernatant was removed and the pellet was washed once with ice-cold PBS (Gibco, Carlsbad, California, USA) and ultracentrifuged again at $100,000 \times g$, 2 hr, 4°C. The resulting pellet was resuspended in 100 µl PBS and transferred to Vivaspin 500 filtration (100,000 MWCO, Sartorius, Göttingen, Germany) for centrifugation at $15,000 \times g$, 45 min. The concentrated EVs were stored at −20 °C until further use for EV characterization.

EV isolation using the precipitation method: 550 µl plasma samples were thawed and first centrifuged at $2000 \times g$ for 20 min. The supernatant was subjected to another round of centrifugation at $10,000 \times g$ for 20 min. After the second round of centrifugation, 500 µl supernatant was mixed with 250 µl PBS, vortexed, and added with 150 µl Exosome Precipitation Reagent. The mixture was incubated at room temperature for 10 min and centrifuged at $10,000 \times g$ for 5 min. After removing the supernatant completely, the pellet was resuspended with 500 µl PBS and concentrated with Vivaspin 500 filtration (100,000 MWCO, Sartorius, Göttingen, Germany) by centrifugation at $15,000 \times g$, 45 min. The resulting EVs were stored at −20°C until further use for EV characterization.

## RNA sequencing

The eluted EV RNAs were first analyzed for their integrity and concentration using Agilent Fragment Analyzer 5200 with DNF-472 High Sensitivity RNA Analysis Kit, 15 nt (Agilent Technologies, Santa Clara, California, United States). A range of 1 ng to 2 µg RNA was used for complementary DNA synthesis as a preparation for EV RNA sequencing (RNA-seq) libraries with SMARTer smRNA-Seq Kit for Illumina (Takara Bio Inc, Mountain View, California, USA) and were sequenced on an Illumina sequencing platform (NextSeq 500/550 Mid Output Kit v2, San Diego, California, USA) with run configurations of single read, read 1:51 cycles, index 1:8 cycles, index 2:8 cycles and an average of 3.7 million reads per sample.

Raw reads were first converted from bcl to fastq format using bcl2fastq (v2.20.0.4.422) and subsequently filtered using FastQ Screen to remove potential contaminations by microorganisms or artifacts due to technical issues. The reads were mapped to a phase II reference genome of the 1000 Genomes Project.

## Radiomics Feature Extraction and different features analysis

We use 3DSlice soft (http://www.slicer.org/) to mark pancreatic benign and PC in CT images as mask and use original figure as reference. Subsequently, we use Python (Version 3.8) software to extract radiomics features, respectively. Afterward, we use the limma package analysis to conduct different feature analysis to identify the significant features between benign and cancer (*Ritchie et al., 2015*). The p-value ≤0.05 was defined as statistically significant.

## Feature selection and radiomics-related signature model build

After obtaining the different features, to avoid multicollinearity of the data, we conduct the dimensionality reduction analysis by Boruta algorithms and LR model. First, the Boruta algorithms will calculate the importance of each feature, and if importance is more than shadow feature, it will be selected as an important feature. Then, input the above important feature into the LR model. The features were selected as significant if the model penalty coefficients were minimized. After selection by machine-learning algorithms, we calculate the regression coefficient of each feature in the LR model. After that, by combining the feature expression to build the signature model in order to predict the images status from large amounts of imaging data. The model formula is listed below. Each regression coefficient of the features is multiplied by its corresponding feature expression level, and then these products are summed. The resulting sum is the risk score. The model validation was tested with applying the WUH center CT dataset, whereas the AUC of the receiver operating characteristic curve was used to evaluate the predictability of the model.

## Definition and identification of low-abundance EV-derived miRNA transcripts

Low-abundance miRNAs are defined as the EV miRNAs with the lowest 30% Counts Per Million (CPM) value across all samples. First, we calculate the CPM value for each miRNA across all samples using the edgeR function (*Robinson et al., 2010*). Then, the non-zero miRNAs with CPM values ranked in the lowest 30% of all samples were selected and defined as low-abundance miRNAs.

## WGCNA analysis to identify imaging feature-related EV miRNAs

According to the radiomics signature, patients will be split into low and high-risk groups. Low-risk classification means images have a high percentage of feature parameters from benign images, while high risk is more likely to become cancer-featuring images. To explore the correlation between the above groups and EV miRNAs, we conduct the WGCNA analysis. The soft-thresholding power of WGCNA was automatically defined by the model, which was then assisting in calculating the expression correlation with (1) a given miRNA to obtain gene significance (GS) and (2) of module membership (MM) with miRNAs to obtain MM. Based on the cut-off criteria (|MM| >0.5 and |GS| >0.1), we obtain the significant miRNAs related to low- and high-risk images, respectively.

## Hub EV miRNAs identify and validate in serum and tissue levels

We merge WGCNA results and low-abundance EV miRNAs to identify the candidate hub miRNAs, then we use GSE109319 to validate this EV miRNAs' different expression between healthy participants and PADC patients in serum level. In addition, we also validate this EV miRNAs different expression between pancreatic tissue of healthy individuals and PADC tissues in TCGA sample cohort.

## Screening the best models to aid hub EV miRNAs in blood-based diagnosis

We collect serum dataset from GSE106817, GSE113486, GSE112264, our hospital center (UMMD), and GSE109319 dataset to perform this procedure. First, the GSE106817 as a training dataset, GSE113486 and GSE112264 are used as independent test datasets. Then we use ten machine-learning algorithms (Gradient Boosting Machine [GBM], KNN, Lasso, XGBoost, ENR, SVM, LR, RR, StepWise, and QDA) composition to select the best model for prediction. After selection based on highest performance values, we apply our hospital (UMMD) dataset and GSE109319 to validate the model accuracy for ability of clinical prediction.

## Common candidate target mRNAs of hub EV miRNAs

To discover the candidate regulation mechanisms of hub EV miRNAs, we use miRPathDB v2.0 database (https://mpd.bioinf.uni-sb.de/overview.html) to predict the candidate target mRNAs of hub EV miRNAs. Subsequently, we select the pass experiment validation mRNAs from the evidence column, as candidate targets of each miRNA. Then, we explore candidate targets of above which were regulated by these three miRNAs at the same time. Finally, we merge the above targets and DHU patient's mRNA sequence to identify the final shared mRNAs which were regulated by three miRNAs at the same time and for the future analysis.

## Cluster of shared target mRNAs and survival analysis between different clusters

We use the R package ConsensusClusterPlus to perform the cluster analysis of common target mRNAs and rank the best cluster results according to the Consensus value output received. Afterward, we also analyze the survival difference between identified subtypes by R package survival. OS and DFS were used as event endpoints.

## Clustering of tumor subtype with clinical factors

To explore the clinical value of each subtype, we analyze the relationship between the subtypes and important clinical factors, such as age, tumor size, and number of positive lymph nodes. We also discover the distribution of sex, PNI, and tumor stage in the different subtypes.

## Characterization of immune cell infiltration properties and immune checkpoint activation in each tumor subtype

We use the MCP-counter algorithms to calculate immune cell infiltration of each sample and calculate the difference in the two tumor subtypes for each immune cell type separately. We also conduct the relationship between the subtypes and immune checkpoint activation to predict the candidate subtype that could benefit from the immune checkpoint inhibitors.

## Potential drug sensitivity for each subtype

We download drug sensitivity data of molecular characterized cell lines to FDA approved, clinical drugs from GDSC database https://www.cancerrxgene.org/, and then use pRRophetic package to estimate the drug sensitivity of the two discovered subtypes.

## Functional enrichment analysis and pathway prediction

To explore the biological function difference in Biological Process, Cellular Component, and Molecular Function, we conduct the Gene Set Enrichment Analysis. We also use this method to analyze the pathway enrichment difference between the two subtypes. Cluster profile package performs this operation and sets p-value ≤0.05 as significant enrichment results.

## Statistics

Radiomics features were extracted and analyzed using the limma package, which applies linear modeling and empirical Bayes moderation suitable for high-dimensional imaging data with multiple comparisons. Feature selection was conducted via the Boruta algorithm to robustly identify all relevant predictors, and an LASSO (Least Absolute Shrinkage and Selection Operator) regression model was built for radiogenomic signature development and validation due to its suitability for high-dimensional and potentially collinear variables. To identify miRNAs correlated with radiomics features, we performed WGCNA, which is appropriate for detecting modules of co-expressed genes across large datasets. Diagnostic performance of candidate miRNAs was assessed using ten different machine-learning classifiers, including random forest and support vector machines, allowing for robust cross-validation. For comparisons between groups (e.g., benign vs. malignant), *t*-tests or non-parametric alternatives were applied depending on data distribution, assessed via Shapiro–Wilk normality test. Survival differences between molecular subtypes were evaluated using Kaplan–Meier analysis with log-rank tests, and multivariate Cox proportional hazards models were used to adjust for confounding clinical variables.

## Materials availability statement

All data generated from this study, if not included in this article, are available from the corresponding authors on reasonable request. The code could be obtained from https://bioconductor.org/packages/release/bioc/html/limma.html.

---

# Additional information

### Funding

| Funder | Grant reference number | Author |
|---|---|---|
| Otto von Guericke University Magdeburg | | Julia Nagelschmitz |
| Research Campus Stimualte, Univeristy of Magdeburg, Germany. | European Regional Development Fund under operation number ZS/2023/12/182010 as part of the initiative 'Sachsen-Anhalt WISSENSCHAFT Schwerpunkte. | Georg Rose Ulf Dietrich Kahlert Wenjie Shi |

The funders had no role in study design, data collection, and interpretation, or the decision to submit the work for publication.

---

## Author contributions
Jianying Xu, Formal analysis, Methodology, Writing – original draft; Wenjie Shi, Data curation, Software, Formal analysis, Investigation, Visualization, Methodology, Writing – original draft; Yi Zhu, Resources, Methodology; Chao Zhang, Writing – original draft, Project administration; Julia Nagelschmitz, Writing – original draft; Maximilian Doelling, Maciej Pech, Resources, Writing – review and editing; Sara Al-Madhi, Conceptualization, Writing – review and editing; Ujjwal Mukund Mahajan, Writing – review and editing; Georg Rose, Funding acquisition, Writing – review and editing; Roland Siegfried Croner, Conceptualization, Resources, Funding acquisition, Writing – review and editing; Guoliang Zheng, Resources, Project administration, Writing – review and editing; Christoph Kahlert, Resources, Methodology, Writing – review and editing; Ulf Dietrich Kahlert, Conceptualization, Resources, Supervision, Funding acquisition, Writing – original draft, Project administration, Writing – review and editing

## Author ORCIDs
Jianying Xu ⓘ http://orcid.org/0009-0007-1924-8400
Wenjie Shi ⓘ https://orcid.org/0009-0001-7345-579X
Christoph Kahlert ⓘ http://orcid.org/0000-0003-4124-7918
Ulf Dietrich Kahlert ⓘ https://orcid.org/0000-0002-6021-1841

## Ethics
The project has been approved by the ethical commission of the Medical Faculty of the University Magdeburg under registry #46/22 and #33/01, respectively. Details are written in the Material and methods section.

Reviewer #1 (Public review): https://doi.org/10.7554/eLife.103737.3.sa1
Reviewer #2 (Public review): https://doi.org/10.7554/eLife.103737.3.sa2
Author response https://doi.org/10.7554/eLife.103737.3.sa3

# Additional files

## Supplementary files
MDAR checklist

## Data availability
All data from public sources can be obtained through the references cited at relevant sections. microRNAseq data from extracellular vesicles from patients' blood plasma can be obtained from NCBI GEO data repository under ascension number GSE304572.

The following dataset was generated:

| Author(s) | Year | Dataset title | Dataset URL | Database and Identifier |
|---|---|---|---|---|
| Prause R, Kahlert C, Kahlert U | 2025 | microRNAseq from extracellular vesicles from blood plasma of pancreatic diseased patients | https://www.ncbi.nlm.nih.gov/geo/query/acc.cgi?acc=GSE304572 | NCBI Gene Expression Omnibus, GSE304572 |

*Continued on next page*

The following previously published datasets were used:

| Author(s) | Year | Dataset title | Dataset URL | Database and Identifier |
|---|---|---|---|---|
| Yokoi Y, Matsuzaki K, Yamamoto Y, Takahashi K, Shimizu H, Uehara T, Ishikawa M, Ikeda S, Kawauchi J, Aoki Y, Niida S, Sakamoto H, Kato K, Kato T, Ochiya T | 2018 | Integrated extracellular microRNA profiling for ovarian cancer screening | https://www.ncbi.nlm.nih.gov/geo/query/acc.cgi?acc=GSE106817 | NCBI Gene Expression Omnibus, GSE106817 |
| Usuba W, Urabe F, Yamamoto Y, Matsuzaki J, Sasaki H, Ichikawa M, Takizawa S, Aoki Y, Niida S, Sakamoto H, Kato K, Egawa S, Chikaraishi T, Fujimoto H, Ochiya T | 2018 | Circulating miRNA profiling in pancreatic cancer vs. healthy controls | https://www.ncbi.nlm.nih.gov/geo/query/acc.cgi?acc=GSE113486 | NCBI Gene Expression Omnibus, GSE113486 |
| Song S | 2019 | Identification of potential biomarkers for diagnosis of pancreatic and biliary tract cancers by sequencing of serum microRNAs | https://www.ncbi.nlm.nih.gov/geo/query/acc.cgi?acc=GSE109319 | NCBI Gene Expression Omnibus, GSE109319 |
| Urabe F, Matsuzaki J, Yamamoto Y, Hara T, Kimura T, Takizawa S, Aoki Y, Niida S, Sakamoto H, Kato K, Egawa S, Fujimoto H, Ochiya T | 2019 | Large-scale and high-confidence serum circulating miRNA biomarker discovery in prostate cancer | https://www.ncbi.nlm.nih.gov/geo/query/acc.cgi?acc=GSE112264 | NCBI Gene Expression Omnibus, GSE112264 |

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
