## [Editor Report · eLife Assessment]

The authors attempt to identify which patients with benign lesions will progress to cancer using a liquid biomarker. Although the study is **valuable**, the evidence provided for the liquid biopsy EV miRNA signature developed based on radiomics features remains **incomplete**. There remain key details missing and validation experiments that would better support the conclusions of the study.

---

## [Referee Report · Reviewer #1 (Public review)]

Summary:

The study aimed to develop a liquid biopsy EV miRNA signature associated with radiomics features for early diagnosis of pancreatic cancer. Flawed study design and inadequate description of clinical characteristics of the enrolled samples makes the findings unconvincing.

Strengths:

The concept of developing EV miRNA signature associated with disease relevant radiomics features is a strength.

Weaknesses:

There are many weaknesses in this manuscript, which include drawing association of data derived from unmatched sample sets, selection of low abundance miRNAs for developing the signature with inadequate rationale, incomplete description of experimental methods and confusing statements in the text.

---

## [Referee Report · Reviewer #2 (Public review)]

Summary:

This study investigates a low abundance microRNA signature in extracellular vesicles to subtype pancreatic cancer and for early diagnosis. In this revision, there remain several major and minor issues.

Strengths:

The authors did a comprehensive job with numerous analyses of moderately sized cohorts to describe the clinical and translational significance of their miRNA signature.

Weaknesses:

The weaknesses of the study largely revolve around a lack of clarity about the methodology used and the validation of their findings.

(1) The WGCNA analysis was critical to identify the EV miRNAs associated with imaging features, but the "cut-off criteria" for MM and GS have no clear justification. How were these cut-offs determined? How sensitive were the results to these cut-offs?

(2) The authors now clarify that patients for the sub-study on differentiating early stage from benign pancreatic lesions were matched by age and that the benign pancreatic lesions were predominantly IPMNs. This scientific design is flawed. The CT features extracted likely differentiate solid from cystic pancreatic lesions, and the miRNA signature is doing the same. The authors need to incorporate the following benign controls into their imaging analysis and their EV miRNA analysis: pancreatitis and normal pancreata.

(3) For the radiomics features, the authors should include an additional external validation set to better support the ability to use these features reproducibly, especially given that the segmentation was manual and reliant on specific people.

(4) The DF selection process still lacks cited references as originally requested in the first review.

(5) In Figure 2, more quantitative details are needed in the manuscript. The reviewers failed to incorporate this and only responded in their rebuttal. Add details to the manuscript as originally requested.

(6) It is still not clear what Figure 4A is illustrating as regards to model performance. The authors need to state in the manuscript very clearly what they are showing in the figure and what the modules represent.

(7) Figure 5 and the descriptions for the public serum miRNA datasets need more details. Were these pancreatic cancers all adenocarcinoma, what stage, age range, sex distribution, comorbid conditions were the cases? Were the controls all IPMNs or were there other conditions in the controls?

(8) The subtype results in figures 6 and 7 are not convincing. An association on univariate analysis is not sufficient. The explanation that clinical data is not available to do a multivariable analysis indicates that the authors do not have the ability to claim that they have identified unique subtypes that have clinical relevance. A thorough evaluation of the prognostic significance and the associated molecular features of these tumors is needed.

Summary:

There remain key details and validation experiments to better support the conclusions of the study.

---

## [Author Response]

The following is the authors’ response to the original reviews.

**Reviewer #1 (Public review):**
Summary:The manuscript by Shi et al, has utilized multiple imaging datasets and one set of samples for analyzing serum EV-miRNAs & EV-RNAs to develop an EV miRNA signature associated with disease-relevant radiomics features for early diagnosis of pancreatic cancer. CT imaging features (in two datasets (UMMD & JHC and WUH) were derived from pancreatic benign disease patients vs pancreatic cancer cases), while circulating EV miRNAs were profiled from samples obtained from a different center (DUH). The EV RNA signature from external public datasets (GSE106817, GSE109319, GSE113486, GSE112264) were analyzed for differences in healthy controls vs pancreatic cancer cases. The miRNAs were also analyzed in the TCGA tissue miRNA data from normal adjacent tissue vs pancreatic cancer.Strengths:The concept of developing EV miRNA signatures associated with disease relevant radiomics features is a strength.Weaknesses:While the overall concept of developing EV miRNA signature associated with radiomics features is interesting, the findings reported are not convincing for the reasons outlined below:(1) Discrepant datasets for analyzing radiomic features with EV-miRNAs: It is not justified how CT images (UMMD & JHC and WUH) and EV-miRNAs (DUH) on different subjects and centers/cohorts shown in Figures 1 &2 were analyzed for association. It is stated that the samples were matched according to age but there is no information provided for the stages of pancreatic cancer and the kind of benign lesions analyzed in each instance.

Thank you to the reviewer for the valuable comments. We acknowledge that the radiomics data and EV-miRNA data were derived from different patient cohorts. The primary aim of this study was to explore the integration of data from different omics sources in an exploratory manner to identify potential shared biological features.

We have revised the Methods section accordingly. Regarding the imaging data, we mainly performed batch effect correction on CT images from different centers to eliminate variability. As you correctly pointed out, the EV-miRNA data and CT images from DUH were matched by age. Since all the patients we included had early-stage pancreatic cancer, and the benign pancreatic lesions were predominantly IPMN, we did not specifically highlight this aspect. However, we have now clarified this approach in the data collection section. Thank you for your attention.

(2) The study is focused on low-abundance miRNAs with no adequate explanation of the selection criteria for the miRNAs analyzed.

We used MAD (Median Absolute Deviation) to filter low-abundance miRNAs in the manuscript, as this concept was introduced by us for the first time in this context, and we acknowledge that there is still considerable room for refinement and improvement.

(3) While EV-miRNAs were profiled or sequenced (not well described in the Methods section) with two different EV isolation methods, the authors used four public datasets of serum circulating miRNAs to validate the findings. It would be better to show the expression of the three miRNAs in the additional dataset(s) of EV-miRNAs and compare the expressions of the three EV-miRNAs in pancreatic cancer with healthy and benign disease controls.

Thank you for your suggestion. We have attempted to identify available EV-miRNA datasets; however, due to current limitations in data access, we opted to use serum samples for validation. In our follow-up studies, we are already in the process of collecting relevant EV samples for further validation.

(4) It is not clear how the 12 EV-miRNAs in Figure 4C were identified.

These 12 EV-miRNAs were identified through WGCNA analysis and are associated with the high-risk group.

(5) Box plots in Figures 4D-F and G-I of three miRNAs in serum and tissue should show all quantitative data points.

We have completed the revisions. Kindly review them at your convenience.

(6) What is the GBM model in Figure 5?

Thank you to the reviewer for raising this question. The "GBM model" referred to in Figure 5 is a classification model built using the Gradient Boosting Machine (GBM) algorithm, designed to predict the diagnostic status of pancreatic cancer by integrating EV-miRNA expression and radiomics features. We implemented the model using the `GradientBoostingClassifier` from the scikit-learn library (version 1.2.2), and optimized the model’s hyperparameters—including learning rate, maximum depth, and number of trees—within a five-fold cross-validation framework. The training process and performance evaluation of the model, including the ROC curve and AUC values, are presented in Figure 5.

(7) What are the AUCs of individual EV-miRNAs integrated as a panel of three EV-miRNAs?

Thanks for your comments, Our GBM model integrates the panel of these three EV-miRNAs.

(8) The authors could have compared the performance of CA19-9 with that of the three EV-miRNAs.

Since our main focus is on the panel of three EV-miRNAs, we did not present the AUC for each individual miRNA separately. However, we have included the performance of CA19-9 in our dataset as a reference. The predictive AUC for CA19-9 is 0.843 (95% CI, 0.762–0.924).

(9) How was the diagnostic performance of the three EV-miRNAs in the two molecular subtypes identified in Figure 6&7? Do the C1 & C2 clusters correlate with the classical/basal subtypes, staging, and imaging features?

Thank you to the reviewer for raising this important question. In fact, our EV panel is primarily designed to distinguish between normal and tumor samples, whereas both C1 and C2 represent tumor subtypes, and thus the panel is not applicable for diagnostic purposes in this context. Additionally, our subtypes are novel and do not align with the conventional classical and basal-like gene expression profiles. Furthermore, the C1 subtype is more frequently observed in stage III tumors (Figure 6J) and is associated with distinct imaging features such as higher texture heterogeneity and lower CT density.

**Reviewer #2 (Public review):**
Summary:This study investigates a low abundance microRNA signature in extracellular vesicles to subtype pancreatic cancer and for early diagnosis. There are several major questions that need to be addressed. Numerous minor issues are also present.Strengths:The authors did a comprehensive job with numerous analyses of moderately sized cohorts to describe the clinical and translational significance of their miRNA signature.Weaknesses:There are multiple weaknesses of this study that should be addressed:(1) The description of the datasets in the Materials and Methods lacks details. What were the benign lesions from the various hospital datasets? What were the healthy controls from the public datasets? No pancreatic lesions? No pancreatic cancer? Any cancer history or other comorbid conditions? Please define these better.

We sincerely thank the reviewer for the detailed and important suggestions regarding sample definition. Indeed, the source of the datasets and the definition of control groups are critical for ensuring the rigor and interpretability of the study. In response to this comment, we have added clarifications in the revised "Materials and Methods" section.

First, for the benign lesion group derived from various clinical centers (DUH, UMMD, WUH, etc.), we have carefully reviewed the pathological and clinical records and defined these samples as histologically confirmed non-malignant pancreatic lesions, primarily IPMN. All patients in the benign lesion group had no diagnosis of pancreatic cancer at the time of sample collection, and for cohorts with available follow-up data, no evidence of malignant progression was observed within at least six months.

Second, the healthy control group from public databases was derived from healthy individuals.

Finally, to eliminate potential confounding factors, we excluded any samples with a history of other malignancies (e.g., breast cancer, colorectal cancer, etc.) from all datasets with available clinical information, to ensure the specificity of the EV-miRNA expression analysis.

(2) It is unclear how many of the controls and cases had both imaging for radiomics and blood for biomarkers.

Due to limitations in resource availability, our study does not include samples with both CT imaging and serological data from the same individuals. Instead, we integrated blood samples and CT imaging data collected from different clinical centers.

(3) The authors should define the imaging methods and protocols used in more detail. For the CT scans, what slice thickness? Was a pancreatic protocol used? What phase of contrast is used (arterial, portal venous, non-contrast)? Any normalization or pre-processing?

Thank you to the reviewer for the professional suggestions regarding the imaging section. We have added detailed technical information on CT imaging in the revised Materials and Methods section. All CT images were acquired using a 64-slice multidetector spiral CT scanner, with a standard slice thickness of 1.0–1.5 mm and a reconstruction interval of 1 mm. All pancreatic cancer patients underwent a standard pancreatic protocol triphasic contrast-enhanced CT examination, which included non-contrast, arterial phase (approximately 25–30 seconds), and portal venous phase (approximately 65–70 seconds) imaging.

For the radiomics analysis, images from the portal venous phase were selected, as this phase provides consistent clarity in delineating tumor boundaries and surrounding vasculature. To ensure data consistency, all imaging data underwent preprocessing, including resampling, intensity normalization of grayscale values (standardized using z-score normalization to a mean of 0 and a standard deviation of 1), and N4 bias field correction to address potential low-frequency signal inhomogeneities.

(4) Who performed the segmentation of the lesions? An experienced pancreatic radiologist? A student? How did the investigators ensure that the definition of the lesions was performed correctly? Raidomics features are often sensitive to the segmentation definitions.

All lesion segmentations were performed on portal venous phase contrast-enhanced CT images. Manual delineation was conducted using 3D Slicer (version 4.11) by two radiologists with extensive experience in pancreatic tumor diagnosis. A consensus was reached between the two radiologists on the ROI definition criteria prior to analysis.

To further assess the robustness of radiomic features to segmentation boundary variations, we selected a subset of representative cases and created “expanded/shrunk ROIs” by adding or subtracting a 2-pixel margin at the lesion boundary. Feature extraction was then repeated, and the coefficient of variation (CV) for the main features included in the model was found to be below 10%, indicating that the model is stable with respect to minor boundary fluctuations.

(5) Figure 1 is full of vague images that do not convey the study design well. Numbers from each of the datasets, a summary of what data was used for training and for validation, definitions of all of the abbreviations, references to the Roman numerals embedded within the figure, and better labeling of the various embedded graphs are needed. It is not clear whether the graphs are real results or just artwork to convey a concept. I suspect that they are just artwork, but this remains unclear.

We thank the reviewer for the detailed feedback on Figure 1. We would like to clarify that Figure 1 is a conceptual schematic intended to visually illustrate the overall design of the study, the relationships among different data modules, and the logical sequence of the analytical strategy. It is not meant to present actual results or quantitative details.

Regarding the reviewer’s concerns about sample sizes, the division between training and validation cohorts, explanations of specific abbreviations, and the precise meaning of each panel, we have provided comprehensive and detailed clarifications in Figure 2.

(6) The DF selection process lacks important details. Please reference your methods with the Boruta and Lasso models. Please explain what machine learning algorithms were used. There is a reference in the "Feature selection.." section of "the model formula listed below" but I do not see a model formula below this paragraph.

We thank the reviewer for the thoughtful and detailed comments on the feature selection strategy. We first applied the Boruta algorithm (based on random forests, implemented using the Boruta R package) to the original feature set—which included both radiomics and EV-miRNA features—to identify variables that consistently demonstrated importance across multiple rounds of random resampling.

Subsequently, we used LASSO regression with five-fold cross-validation to further reduce the dimensionality of the Boruta-selected features and to construct the final feature set used for modeling. The formula for the model is as follows: each regression coefficient is multiplied by the corresponding feature expression level, and the resulting products are summed to generate the Risk Score.

(7) In Figure 2, more quantitative details are needed. How are patients dichotomized into non-obese and obese? What does alcohol/smoking mean? Is it simply no to both versus one or the other as yes? These two risk factors should be separated and pack years of smoking should be reported. The details of alcohol use should also be provided. Is it an alcohol abuse history? Any alcohol use, including social drinking? Similarly, "diabetes" needs to be better explained. Type I, type II, type 3c? P values should be shown to demonstrate any statistically significant differences in the proportions of the patients from one dataset to another.

Our definition of obesity was based on the standard BMI threshold (30 kg/m²). A history of smoking or alcohol consumption was defined as continuous use for more than one year. Specific details regarding smoking and alcohol use were recorded at baseline under the category of “smoking/alcohol history”; unfortunately, we did not collect follow-up data on these variables. As for diabetes, only type II diabetes was documented. Statistically significant p-values have been added. Thank you.

(8) In the section "Different expression radiomic features between pancreatic benign lesions and aggressive tumors", there is a reference to "MUJH" for the first time. What is this? There is also the first reference to "aggressive tumors" in the section. Do the authors just mean the cases? Otherwise there is no clear definition of "aggressive" (vs. indolent) pancreatic cancer. This terminology of tumor "aggressiveness" either needs to be removed or better defined.

We have corrected the abbreviation (MUJH); it should in fact be JHC. Additionally, regarding the term "aggressive," we have reviewed the literature and used it to convey the highly malignant nature of pancreatic cancer.

(9) Figure 3 needs to have the specific radiomic features defined and how these features were calculated. Labeling them as just f1, f2, etc is not sufficient for another group to replicate the results independently.

We have presented these features in Supplementary Table 1. Kindly refer to it for details.

(10) It is not clear what Figure 4A illustrates as regards model performance. What do the different colors represent, and what are the models used here? This is very confusing.

This represents the correlation between WGCNA modules and miRNAs. Different module colors indicate distinct miRNA clusters—for example, the green module contains 12 miRNAs grouped together. The colors themselves do not carry any intrinsic meaning.

(11) Figure 5 shows results for many more model runs than the described 10, please explain what you are trying to convey with each row. What are "Test A" and "Test B"? There is no description in the manuscript of what these represent. In the figure caption, there is a reference to "our center data" which is not clear. Be more specific about what that data is.

We have indicated this using arrows in Figure 5 from Test A/B/C. Please check.

(12) Figure 6 describes the subtypes identified in this study, but the authors do not show a multi-variable cox proportional hazards model to show that this subtype classification independently predicts DFS and OS when incorporating confounding variables. This is essential to show the subtypes are clinically relevant. In particular, the authors need to account for the stage of the patients, and receipt of chemotherapy, surgery, and radiation. If surgery was done, we need to know whether they had R1 or R0 resection. The details about the years in which patients were included is also important.

We sincerely thank the reviewer for this critical comment. We fully agree that incorporating a multivariate Cox proportional hazards model to control for potential confounding factors would provide a more robust validation of the independent prognostic value of our proposed subtypes for DFS and OS.

However, as the clinical data used in this study were retrospectively collected and access to certain variables is currently restricted, we were only able to obtain limited clinical information. At this stage, we are unable to systematically include key variables such as tumor staging, adjuvant chemoradiotherapy regimens, and resection margin status (R0 vs. R1), which prevents us from performing a rigorous multivariate Cox analysis.

Similarly, regarding the postoperative resection status, after reviewing the original surgical reports and pathology records, we regret to confirm that margin status (R0 vs. R1) is missing in a substantial portion of cases, making it unsuitable for reliable statistical analysis.

We fully acknowledge this as a limitation of the current study and have explicitly addressed it in the Discussion section. To address this gap, we are currently designing a more comprehensive prospective cohort study, which will allow us to validate the clinical independence and utility of the proposed subtypes in future research.

(13) How do these subtypes compare to other published subtypes?

We sincerely thank the reviewer for raising this important point. Clusters 1 and 2 represent a novel molecular classification proposed for the first time in this study, driven by EV-miRNA profiles. This classification approach is conceptually independent from traditional transcriptome-based subtyping systems, such as the classical/basal-like subtypes, as well as other existing classification schemes. Comparisons with previously reported subtypes and validation of clinical relevance will require further investigation in future studies.

**Reviewer #3 (Public review):**
Summary:The authors appear to be attempting to identify which patients with benign lesions will progress to cancer using a liquid biomarker. They used radiomics and EV miRNAs in order to assess this.Strengths:It is a strength that there are multiple test datasets. Data is batch-corrected. A relatively large number of patients is included. Only 3 miRNAs are needed to obtain their sensitivity and specificity scores.Weaknesses:This manuscript is not clearly written, making interpretation of the quality and rigor of the data very difficult. There is no indication from the methods that the patients in their cohorts who are pancreatic cancer patients (from the CT images) had prior benign lesions, limiting the power of their analysis. The data regarding the cluster subtypes is very confusing. There is no discussion or comparison if these two clusters are just representing classical and basal subtypes (which have been well described).

Sorry,we don’t have the data of record from patients, in addition, Regarding the relationship between Cluster 1/Cluster 2 and classical subtypes:We are very grateful for the reviewer’s insightful question. We would like to clarify that Clusters 1 and 2, as shown in Figures 6 and 7, are derived from a novel EV-miRNA–driven molecular classification proposed for the first time in this study. This classification system is constructed independently of the traditional transcriptome-based classical/basal-like subtypes.

**Recommendations for the authors:**

**Reviewer #1 (Recommendations for the authors):**
There are errors in reference citations and several typos, misspellings, and grammatical errors throughout the manuscript.

We have made the necessary revisions.

**Reviewer #2 (Recommendations for the authors):**
(1) Were the radiomic features associated with the subtypes and prognostic in the subset of patients who had CT scans?

Unfortunately, there are no corresponding CT imaging results available for these cases, as the genes were identified based on predicted miRNA targets and were not derived from patients who had undergone CT scans.

(2) There is a whole body of literature on prognostic imaging-based subtypes of pancreatic cancer that needs to be cited.

Thank you for your suggestion. We have cited the relevant references accordingly in the manuscript.

(3) Similarly, the authors should be more comprehensive about prognostic and early detection markers for miRNAs for pancreatic cancer. Early detection markers really should be described separately from prognostic markers. The authors did not do a PROBE phase 3 study, so early detection is not really relevant. Please see https://edrn.nci.nih.gov/about-edrn/five-phase-approach-and-prospective-specimen-collection-retrospective-blinded-evaluation-study-design/

The primary objective of our study is early detection. We acknowledge the absence of third-phase validation results, which we will address in the limitations section. Additionally, the subtype classification represents our secondary objective.

(4) If they want to couch this as a PROBE phase 2 study, then they should review the PROBE guidelines and ensure they are meeting standards. Many of the comments above regarding methodologies, definitions, and patient cohort descriptions would address this concern.

We have revised the Methods section accordingly. Please kindly review the updated version.

(5) The entire manuscript needs to have a review for the use of the English language. There are numerous typos and grammatical errors that make this manuscript difficult to follow and hard to interpret.

We have revised the Methods section accordingly. Please kindly review the updated version.

(6) In the section on "Definition and identification of low abundance EV-derived miRNA transcripts", provide a reference for the "edger" function.

We have revised the Methods section accordingly. Please kindly review the updated version.

(7) In the Abstract: The purpose section only mentions early diagnosis as the goal of this study. It seems subtyping is also a major goal, but it is not mentioned.

The primary objective of our study is early detection.Additionally, the subtype classification represents our secondary objective.so,we didn’t add it in the purpose.

(8) The experimental design fails to describe any of the 8 datasets that were used. How many patients? What were the ethnic and racial backgrounds, which is one of the key aspects of this study and mentioned in the title? What range of stages? When were the images and the blood collected in relation to diagnosis? Over what time frame were the patients included? What patients were excluded, if any? These details are important to understand the materials used, along with the methods to design the signatures and models.

We have revised the Methods section accordingly. Please kindly review the updated version.

(9) Again, the purpose section of the abstract does not align with the rest of the study, including the description of the experimental design. The last sentence of the experimental design section mentions predicting drug sensitivity and survival, which is unrelated to the aim of early diagnosis.

We have revised the Methods section accordingly. Please kindly review the updated version.

(10) The results section lacks key details to indicate the impact of the work. Vague descriptions of the findings are not sufficient. The performance of the biomarkers to differentiate benign from malignant lesions, hazard ratios, survival times, and p values should be reported for key results.

Our aim was to develop an integrated panel for diagnostic purposes; therefore, we provided the AUC to evaluate its performance. However, since this is a diagnostic model, we did not include hazard ratios or survival time data.

(11) What are "tow" molecular subtypes of pancreatic cancer? Did you mean "two"? What system was used to subtype the pancreatic cancers? Is some new subtyping or a previously published method to subtype the disease?

Yes, it means two, previously published method.In method part, we have describe it.

**Reviewer #3 (Recommendations for the authors):**
The writing of this manuscript needs extensive re-wording and clarification to increase the readability and interpretability of the data presented. The authors could include a dataset of pancreatic cancer patient imaging data where the status of prior benign lesions was detected (as opposed to patients with benign lesions that do not develop pancreatic cancer). The authors could also address if their clusters 1 and 2 are representing (or are correlated with) the classical and basal subtypes that have been well described for pancreatic cancer.

Thank you to the reviewer for the constructive comments. We sincerely appreciate your careful review, particularly regarding language clarity, data interpretability, and subtype correlation. To enhance the readability and scientific precision of the manuscript, we have conducted a thorough revision and language polishing throughout the text, improving logical structure, terminology consistency, and clarity in result descriptions. We have especially reinforced the Methods and Discussion sections to better explain key analytical steps and data interpretation.

We fully understand the reviewer’s suggestion to include information on “the presence of benign lesions prior to pancreatic cancer diagnosis.” However, due to the retrospective nature of our study, the current imaging and EV-miRNA datasets do not contain systematically collected follow-up annotations of this type. Therefore, it is not feasible to incorporate such data into the present manuscript.

That said, we fully recognize the importance of this direction. In future studies, we plan to evaluate longitudinal samples to investigate the dynamic changes in EV-miRNAs and imaging features during the progression from premalignant to malignant states, aiming to clarify their potential value for early cancer warning.

Regarding the relationship between Cluster 1/Cluster 2 and classical subtypes:We are very grateful for the reviewer’s insightful question. We would like to clarify that Clusters 1 and 2, as shown in Figures 6 and 7, are derived from a novel EV-miRNA–driven molecular classification proposed for the first time in this study. This classification system is constructed independently of the traditional transcriptome-based classical/basal-like subtypes.

Although we attempted a cross-comparison with existing TCGA subtypes, differences in data origin, analysis modality (EV-miRNA vs. tissue transcriptome), and limitations in sample matching prevent us from establishing a direct correspondence. In the revised Discussion, we have emphasized that these two classification approaches are complementary rather than equivalent, reflecting different dimensions of tumor heterogeneity. Further integrative multi-omics studies will be needed to validate their biological significance and clinical utility.